# The General Hopelessness Scale: Development of a measure of hopelessness for non-clinical samples

**Ken Drinkwater**[1]*, **Andrew Denovan**[2], **Neil Dagnall**[1], **Chris Williams**[1]

**1** Department of Psychology, Manchester Metropolitan University, Manchester, United Kingdom,
**2** Department of People and Performance, Manchester Metropolitan University, Manchester, United Kingdom

* k.drinkwater@mmu.ac.uk

**Data Availability Statement:** All data files are available from figshare: https://figshare.com/s/ cf5ae31bcc0b04d94b91 https://figshare.com/s/ 91e7372ea710191a9d90.

## Abstract

Noting concerns about the non-clinical efficacy of the Beck Hopelessness Scale (BHS), specifically the instrument's ability to discriminate between lower levels of hopelessness, this paper describes the development of the General Hopelessness Scale (GHS) for use with general samples. Following a literature review an item pool assessing the breadth of the hopelessness construct domain was created. This was then placed in survey form and assessed within two independent studies. Study 1 ($N$ = 305, 172 women, 133 men, $M$age = 28.68) explored factorial structure, item performance, and convergent validity of the GHS in relation to standardised measures of self-esteem and trait hopelessness. In Study 2 ($N$ = 326, 224 women, 102 men, $M$age = 26.52), scrutiny of the GHS occurred using confirmatory factor analysis and invariance tests, alongside item performance and convergent validity analyses relative to measures of affect, optimism, and hope. Factor analysis (using minimum average partial correlations and exploratory factor analysis) within Study 1 revealed the existence of four dimensions (Negative Expectations, Hope, Social Comparison, and Futility), which met Rasch model assumptions (i.e., good item/person fit and item/person reliability). Further psychometric assessment within Study 2 found satisfactory model fit and gender invariance. Convergent validity testing revealed moderate to large associations between the GHS and theoretically relevant variables (self-esteem, trait hopelessness, affect, optimism, and hope) across Study 1 and 2. Further examination of performance (reliability and ceiling and floor effects) within Study 1 and 2 demonstrated that the GHS was a satisfactory measure in non-clinical settings. Additionally, unlike the BHS, the GHS does not assume that administrators are trained professionals capable of advising on appropriate interventions.

## Introduction

This paper validated the General Hopelessness Scale (GHS). This novel, theoretically informed measure was produced to assess hopelessness in general populations. The advancement of the GHS was necessary because the current predominant measure, Beck's Hopelessness Scale (BHS)

**Funding:** The author(s) received no specific funding for this work.

**Competing interests:** The authors have declared that no competing interests exist.

(BHS [1]), originated from work with clinical samples. Accordingly, critics assert that the BHS is not able to effectively distinguish between lower levels of hopelessness [2]. Acknowledging this concern and the fact that non-clinical populations typically possess relatively low levels of hopelessness the authors developed the GHS.

Hopelessness denotes the tendency to overestimate adverse events and underestimate the probability of positive occurrences [3]. This is characterised typically as negative expectation or pessimistic attitude toward self and/or future [4]. Hence, hopelessness expresses as low expectation of goal achievement, reduced belief in probability of success, and feelings of futility [5]. Furthermore, theorists make a distinction between levels of state and trait hopelessness [6]. Trait represents dispositional baseline, whereas state varies as a function of contextual and situation factors (i.e., stressors and mood episode).

Hopelessness is an important construct because it is a principal characteristic of depression and an important etiological and maintaining factor for suicide risk [7–9]. Correspondingly, hopelessness is an integral element of theoretical models of suicide [10]. In this context, the capacity to detect subtle differences at the lower end of the score distribution is important since even moderate levels of hopelessness are indicative of low subjective well-being and psychological difficulties.

## Beck Hopelessness Scale (BHS)

The most used measure is BHS [1, 11]. The importance of the instrument is demonstrated by its extensive use in published research and its translation into numerous languages (e.g., Nigerian [12], Urdu [13], Hungarian [14]). Psychometric validation of the BHS identified three factors tapping affect, motivation, and cognition [1]. However, ensuing scrutiny has questioned the validity of this factorial structure [15]. Hence, though several studies have replicated the original model [16–19], others report alternatives solutions comprising one [20–23] to five dimensions [24].

The unidimensional model comprises a single underlying factor with positive and negative item phrasing effects [20–23], whereas two-factor structures reflect negative expectation of the future and perceptions of powerlessness [25–27]. More complex models identify multiple dimensions [24, 28]. Illustratively, Nekanda-Trepka, Bishop, and Blackburn [28] identified five factors (Motivation and Outcome Expectation, Confidence in the Future, Future Accomplishment, Trust in the Future, and Time Perspective), and Zhang et al. [24] reported a five-factor model for non-suicide attempters (Feelings About the Future, Pessimistic Motivation, Positive Expectation, Negative Expectation, and Future Expectation), and a four-factor model for suicide attempters (Loss of Motivation, Positive Expectation, Negative Expectation, and Future Expectation).

Noting debate about the underlying factorial structure of the BHS, Boduszek and Dhingra [15] tested a range of models using a large student sample ($N = 1,733$). Analysis revealed that a three-factor solution with method effects provided superior fit. This study was important because it supports the solution proposed by Beck et al. [1] within a non-clinical sample. However, other studies using student samples have failed to reproduce the solution [29, 30]. Moreover Steed [30], following comparisons of BHS with Hope Scale (HS) and Life Orientation Test scores (LOT), concluded that because the HS and LOT were developed to index healthy characteristics and behaviours, they had greater applicability to normal populations than the BHS.

Other authors also note limitations when using the BHS with non-clinical populations. Illustratively, Young et al. [2] observed that the BHS was an inefficient instrument for samples in which the level of hopelessness was expected to be low. Indeed, the BHS works optimally

with higher levels of hopelessness, although it can detect changes down to moderately low levels. Acknowledging this, even in clinical samples the BHS may be less sensitive to important changes in depression within the lower range (i.e., patients progressing from improved to totally remitted).

It is important to identify differences within the medium-lower range because scores potentially reveal subsequent psychological states. Thus, the ability to reliably detect differences across the score range, both between groups and within respondents, is crucial. Despite this, recent studies with university-based samples have continued to use the BHS. Adoption of the BHS is typically predicated on the fact that the instrument is generally regarded as the predominant measure of hopelessness [31–34]. Concerning clinical samples, researchers have also noted variations in the predictive validity of the BHS. For example, Niméus, Träskman-Bendz, and Alsén [35] reported that the BHS failed to significantly predict future suicide in hospitalized suicide attempters. Collectively, these findings imply that the BHS is most effective when it is indexing depression ratings, mood disorders and/or personality disorders [35].

## Conceptualisation

Noting conceptual concerns about the BHS's factorial structure and possible non-clinical efficacy, this paper developed an alternative hopelessness measure for use with general, non-clinical populations. That is, for testing within samples where lower levels of hopelessness are anticipated. To ensure that the new instrument appropriately indexed the construct, the authors incorporated core elements from the delineations of Beck and the theory of hopelessness depression [36].

Both approaches represent cognitive-vulnerability-stress models as they centre on the notion that hopelessness develops from negative expectations of future events [37]. In addition, they emphasise the propensity to negative thinking styles, which Beck refers to as dysfunctional attitudes, and Abramson et al. [36] denotes as negative inferences. The interaction between cognitive styles and negative events is theoretically salient through its predictive capacity for depressive symptoms [38].

Furthermore, both explain why 'some' individuals develop hopelessness and depression, and others do not. Despite similarities there are important differences between the models. Notably, Beck et al. [1] posits that vulnerability to depression arises from an individual's universal beliefs and rules for happiness (i.e., those related to perfectionism, performance, and self-worth), whereas Abramson et al. [36] attributed susceptibility to inferences about the cause, consequences, and self-implications of negative life events [39].

Previously, few studies have tried to integrate these accounts. One notable attempt by Pössel and Thomas [40], explored how elements of the cognitive triad assimilated into the model of hopelessness depression. Since mediation occurred only when the whole cognitive triad was included, this was only partially successful. A further attempt by Pössel and Knopf [37] found that inference style was distinct from the depressogenic schemata and cognitive errors element of Beck's model. Additionally, Pössel and Smith [41] provided support for an integrated model when inferential styles were located between the cognitive errors and cognitive triad. This demonstrated that it was possible to meaningfully combine theoretical accounts.

## The present study

To address BHS limitations (i.e., structural variations as a function of sample type and lack of measurement sensitivity within non-clinical samples [16–17, 26]) this paper outlines the development and validation of the General Hopelessness Scale (GHS). The GHS is intended for groups where low scores are anticipated. Although other scales exist, these like the BHS, were

designed for use within clinical settings/contexts (i.e., State-Trait Hopelessness Scale [42]; Brief Inventory for Helplessness, Hopelessness and Haplessness [43]; and Hopelessness Scale for Children [44]). Also, they lack the conceptual basis of the BHS. Accordingly, the GHS was developed within a non-clinical sample and scale items derived from core elements of the BHS [1] and Abramsons's theory [36]. The Hopelessness Depression Symptom Questionnaire (HDSQ, [45]) served also as a reference point for emerging symptoms. The HDSQ is 32-item self-report measure of symptoms that comprise hopelessness depression [36]. The scale serves as a tool for testing the hopelessness theory of depression and acts as a means for assessing individual symptoms in clinical work and research [45].

Acknowledging these theoretically important sources, GHS items index 'negative perceptions of future events' (e.g., "If something bad could happen, it probably will") and the 'self and one's own adequacies', (e.g., "If I don't get something right after a few tries, I'll probably never get it right"). Items reference also specific features from cognitive models. Explicitly, 'attributions for adverse events and consequences' (e.g., "I often find things to be out of my control"). These reflect the notion that an individual is incapable of taking control of a situation, or that adverse events arise from dispositional factors.

In combination, these components reflect the stability and globality of negative conditions, which Abramson et al. [36] identified as integral to the development of hopelessness. Consistent with this, these items infer or draw normative comparisons with others (e.g., "I doubt I'll get the things that other people have"). This is reflective of the cognitive distortions of cognitive theory as well as the inferred negative consequences and consensus elements of hopelessness depression theory. Finally, the GHS contains statements with positive valance, which are indicative of a confident outlook (e.g., "I look forward to the future with optimism"). These reflect the hopefulness evident in BHS and benign HDSQ statements.

To test the GHS, the present investigation utilised an established procedure of initially exploring factor structure and item performance (Study 1), before testing the stability of the latent structure with an independent sample (Study 2). Moreover, Study 1 contained the GHS alongside pre-established measures of hopelessness. Study 2 contained measures assessing individuals on adaptive qualities, such as positive affect and hopefulness. This method ensured that the GHS was an accurate and robust index of hopelessness.

## Study 1 Materials and methods

### Participants

Study 1 comprised 305 participants (172 women, 56%; 133 men, 44%), mean age 28.68 ($SD$ = 10.82), range 18 to 70. These originated mainly from the USA and UK (41% and 25% respectively) (Table 1). Regarding vocational status, 39% of the sample were students, 54% employed and 7% unemployed. Data screening revealed acceptable univariate skewness and kurtosis for study variables (i.e., between -2.0 to +2.0). In addition, non-normality existed, as Mardia's [46] kurtosis ($b2p$ = 26.19, $p < 0.001$) and skewness ($b1p$ = 162.19, $p < 0.001$) inferred significant deviation.

### Measures

**BHS instrument development.** To create a scale with satisfactory content validity, 31 questions were constructed following examination of extant measures, including the BHS [1] and HDSQ [45]. The semantic content of these scales acted as theoretical reference points to accompany abstract ideas arising from concepts within the cognitive triad theory of depression and the theory of hopelessness depression.

Table 1. Sample characteristics.

| Characteristic | Study 1 (*N* = 305) | | Study 2 (*N* = 326) | |
| --- | --- | --- | --- | --- |
| | % | (*n*) | % | (*n*) |
| Gender | | | | |
| Men | 43 | 133 | 69 | 224 |
| Women | 57 | 174 | 31 | 102 |
| Nationality | | | | |
| UK | 25 | 76 | 34 | 109 |
| USA | 41 | 125 | 23 | 75 |
| Europe (other than UK) | 11 | 35 | 14 | 47 |
| Other | 23 | 69 | 29 | 95 |
| Occupation | | | | |
| Student | 39 | 120 | 74 | 246 |
| Employed | 54 | 164 | 24 | 79 |
| Unemployed | 7 | 21 | 2 | 6 |

Seventeen items indexed negative expectations of future events (9-items) as well as perceptions of inferiority when comparing the self to others (8-items). Eight items assessed general lack of motivation including unwillingness to proceed (e.g., *"I don't want to do X"*) and cognitive dismissal of the outcome (e.g., *"There's no point to doing X"*). A further six items referenced hopeful belief and were reverse-keyed. The measure utilised a 7-point Likert scale (1 = *Strongly disagree* to 7 = *Strongly agree*).

To ensure item comprehension and face validity the initial iteration of the GHS was reviewed by academics experienced in psychometrics. In addition to the suggestion of minor grammatical and semantic alterations, this process recommended a reworking of an item related to comparing the self to others; the item did not fully reflect other associated statements. Following revisions, the reviewers confirmed that the GHS was suitable for psychometric evaluation. To assess concurrent validity, Study 1 participants completed the Trait element of the State-Trait Hopelessness Scale [42] and the Rosenberg Self-Esteem Scale (RSE) [47].

**The Trait element of the State-Trait Hopelessness Scale (THS).** The THS consists of 13 items assessing beliefs and feelings associated with hopelessness, using the phrase 'typically' to represent trait hopelessness. Participants answered using a 7-point Likert scale (*1 = Strongly agree*, *7 = Strongly disagree*). Reported reliability is high for the measure ($\alpha$ = .91 [48]). In this study, good alpha and omega reliability existed ($\alpha$ = .93, $\omega$ = .93).

**The Rosenberg Self-Esteem Scale (RSE).** The RSE is a 10-item measure which uses a 4-point Likert scale (*0 = Strongly disagree*, *3 = Strongly agree*) to assess global self-esteem. The RSE demonstrates excellent internal consistency ($\alpha$ = .92 [47]) with this study showing similarly high reliability ($\alpha$ = .93, $\omega$ = .93).

## Procedure

Prior to participation, respondents read the study brief. This contained background information about the study and outlined the conditions and requirements of involvement. To participate, respondents provided informed consent and were required to be over 18 years of age. This was the only inclusion criteria. Acknowledgement of consent involved respondents ticking/clicking a box within the online survey hosted by Qualtrics (a web-based online survey tool) indicating that they understood the nature of the study and intended to participate. They were informed that they could cease participation at any point during completion of the

measures. Only consenting participants progressed to the online measures. Further instructions asked participants to take their time, complete all questions, and answer openly/honestly. The initial questions requested demographic details (i.e., age, preferred gender, and occupation).

## Ethics

The research team gained ethical approval for the project (Developing the General Hopelessness Scale) from the relevant governing/institutional review board (specific details withheld for anonymity).

## Analytical strategy

Assessment of the General Hopelessness Scale (GHS) advanced through a series of analytical stages. These involved an initial test of factor structure (MPlus Version 8.6; [49]) via parallel and exploratory factor analysis (EFA), which utilised two criteria for factor extraction: Velicer's minimum average partial (MAP) test, and an eigenvalue equal to or greater than 1. Velicer's MAP test computes partial correlations using the covariances among the residuals. Computation of the average squared partial correlation occurs, and the test terminates when this value achieves a minimum result indicating no additional common variance extracted. This is an empirically supported approach for establishing the quantity of factors underlying a measure [50]. Next, Rasch analysis (Winsteps) assessed infit and outfit (MNSQ) for each item, item and person reliability, and item and person separation index. Differential item functioning (DIF) tested response bias among men and women.

Given the presence of non-normality, and ordinal nature of the data, EFA was performed on a polychoric correlation matrix using mean and variance adjusted weighted least squares (WLSMV) estimation. Oblique rotation was applied because correlations among factors were anticipated [51]. Model fit interpretation included chi-square ($\chi^2$), Comparative Fit Index (CFI), Root-Mean-Square Error of Approximation (RMSEA), and Standardised Root-Mean-Square Residual (SRMR). RMSEA scrutiny involved reference to its 90% confidence interval (CI). Fit was determined using traditional cut-offs as indicated by the literature [52, 53]. Explicitly, CFI > 0.90, RMSEA < .10, and SRMR < .08 implied satisfactory fit.

Concurrent validity included correlating the GHS with the THS and RSE scales. Cohen's criteria [54] facilitated interpretation of correlation strength. Specifically, 0.1–0.29 indicated a weak correlation; 0.3–0.49 suggested a moderate relationship; and 0.50 or greater inferred a strong correlation. Cronbach's alpha and coefficient omega ($\omega$) examined internal consistency of the GHS. Analysis subsequently considered ceiling and floor effects.

## Study 1 Results

**Exploratory Factor Analysis (EFA).** The MAP test for Study 1 suggested extraction of four factors, which was consistent with the a priori conceptualisation of items into domains of negative expectations, perceptions of inferiority, lack of motivation and an element of hopefulness. The smallest average squared partial correlation was .01, occurring alongside four underlying components. Further assessment revealed satisfactory sampling adequacy; Kaiser-Meyer-Olkin measure (KMO) = .96 and a reasonable item matrix, Bartlett's Test of Sphericity ($p < 0.001$).

Consistent with the MAP results, EFA revealed four dimensions possessed eigenvalues > 1. Moreover, the four-factor model revealed satisfactory fit across indices, $\chi^2$ (347) = 863.23, $p <$ .001, CFI = .97, RMSEA = .07 (90% CI of .06 to .07), SRMR = .03. Six items loaded below 0.43 [55]. Reanalysis following the removal of these items revealed a more parsimonious solution

**Table 2. Factor loadings of the General Hopelessness Scale items.**

| Item | Factor | | | |
| --- | --- | --- | --- | --- |
| | Hope | Negative Expectations | Social Comparison | Futility |
| 1 | **.64** | -.02 | -.04 | -.13 |
| 4 | **.62** | -.21 | -.10 | .05 |
| 21 | **.67** | .01 | .15 | -.18 |
| 22 | **.90** | -.03 | -.08 | .08 |
| 24 | **.94** | -.03 | -.07 | .11 |
| 25 | **.63** | .02 | .01 | -.13 |
| 2 | -.02 | **.89** | -.02 | .05 |
| 3 | -.16 | **.79** | -.11 | .14 |
| 6 | -.26 | **.51** | .08 | .02 |
| 7 | -.12 | **.45** | .34 | -.11 |
| 8 | .05 | **.62** | .22 | -.06 |
| 15 | -.29 | **.50** | .04 | .16 |
| 16 | .02 | **.88** | .06 | -.07 |
| 23 | .01 | **.63** | .26 | ,02 |
| 5 | .38 | .05 | **-.50** | -.06 |
| 9 | -.31 | .17 | **.56** | -.02 |
| 11 | .12 | .22 | **.62** | .01 |
| 12 | .05 | -.04 | **.75** | .04 |
| 13 | -.05 | -.01 | **.84** | -.02 |
| 17 | -.03 | .16 | **.62** | .06 |
| 18 | -.17 | -.01 | **.76** | .02 |
| 19 | .01 | .11 | **.55** | .25 |
| 10 | -.12 | .01 | .37 | **.45** |
| 14 | .02 | -.02 | .35 | **.75** |
| 20 | -.11 | .17 | .22 | **.46** |

*Note*. Bold values emphasise the loads of items on the corresponding factor, to distinguish from other cross-loads

(i.e., CFI increase of .003, SRMR reduction of .003), $\chi^2$ (206) = 569.52, $p < .001$, CFI = .97, RMSEA = .07 (90% CI of .06 to .07), SRMR = .03. Eight items loaded on Factor 1, six on Factor 2, eight on Factor 3, and three on Factor 4 (Table 2). Factor 1 (labelled as 'Social Comparison'; Eigenvalue of 12.69) comprised items relating to how someone perceives themselves in an interpersonal spectrum, with hopeless individuals likely perceiving themselves as inferior to others.

Items informing Factor 2 consisted mostly of reversed items and comprised positive valency (named 'Hope', Eigenvalue of 2.31). This is like Factor 1 of the Beck Hopelessness Scale, which a user would typically answer as 'false' if they were experiencing levels of hopelessness. The third factor (labelled 'Negative Expectations', Eigenvalue of 1.32) referred to adverse feelings that individuals have regarding their future. The fourth factor consisted of items that capture an individual's aversion to initiate or maintain new behaviours (named 'Futility', Eigenvalue of 1.01).

All four factors evidenced moderate to large inter-factor correlations. Specifically, Social Comparison demonstrated a large association with Hope ($r$ of -.42), Negative Expectation ($r$ of .57), and Futility ($r$ of .30). Hope evidenced a large negative association with Negative Expectation ($r$ of -.67) and Futility ($r$ of -.42). Negative Expectation also correlated strongly with Futility ($r$ of .37).

**Table 3. Psychometric properties of the General Hopelessness Scale at the item level.**

| Item | Factor | Infit MNSQ | Outfit MNSQ | Difficulty | DIF contrast across gender |
|------|--------|-----------|------------|-----------|----------------------------|
| 1. I know I can accomplish what I'm trying to do (R) | Hope | .76 | .86 | -.14 | -.06 |
| *2. Things typically don't work out for me* | Negative Expectations | .89 | .58 | .58 | - |
| 3. I don't see things ever going my way | Negative Expectations | .78 | .79 | .44 | -.11 |
| 4. If I try hard enough, I can get what I want (R) | Hope | 1.13 | 1.13 | -.26 | -.11 |
| *5. When bad things happen, I can easily pick myself back up (R)* | Social Comparison | -.50 | 2.76 | 4.11 | - |
| 6. I doubt I'll get the things that other people have | Negative Expectations | 1.15 | 1.14 | -.25 | -.19 |
| 7. I often find things to be out of my control | Negative Expectations | 1.07 | 1.13 | -.67 | .16 |
| 8. If something bad could happen, it probably will | Negative Expectations | 1.16 | 1.20 | -.13 | .01 |
| 9. Sometimes, everything seems pointless | Social Comparison | 1.01 | .96 | .02 | -.16 |
| 10. I feel that giving up is easier than failing | Futility | 1.09 | 1.00 | -.40 | -.25 |
| 11. I know that others notice my failings | Social Comparison | 1.03 | 1.12 | .25 | .01 |
| 12. I struggle to focus on all that I have to do | Social Comparison | 1.24 | 1.40 | -.02 | -.38 |
| 13. When things go wrong, I start to feel depressed | Social Comparison | .68 | .73 | -.28 | .07 |
| 14. I avoid attempting new things in case I find them difficult | Futility | .83 | .82 | .08 | -.02 |
| 15. I can't imagine getting what I want | Negative Expectations | 1.06 | 1.08 | .26 | .10 |
| 16. I usually have more bad things happen than good | Negative Expectations | .81 | .79 | .26 | -.12 |
| 17. I worry that new people will quickly notice my shortcomings | Social Comparison | .76 | .73 | .10 | .11 |
| 18. I can't find the energy to do the things I need to do | Social Comparison | .69 | .71 | -.06 | .01 |
| 19. I get nervous performing tasks when other people are there | Social Comparison | .94 | .91 | -.18 | .31 |
| 20. If I don't get something right after a few tries, I'll probably never get it right | Futility | 1.04 | 1.00 | .32 | .30 |
| 21. I believe I can make things better for people (R) | Hope | 1.28 | 1.34 | -.46 | -.19 |
| 22. I look forward to the future with optimism (R) | Hope | .75 | .76 | .53 | .13 |
| 23. Things always seem to happen that stop me from getting anywhere | Negative Expectations | .94 | .93 | .09 | .15 |
| 24. I am hopeful about the future (R) | Hope | .66 | .67 | .33 | .20 |
| *25. Seeing other people's successes inspires me to be better (R)* | Hope | 1.30 | 1.44 | - | - |

Items in italics removed following Rasch analysis. (R) denotes reverse-keyed item.

**Rasch analysis.** Each factor was subjected to Rasch analysis. The properties of most items were satisfactory. However, three items possessed unsatisfactory Infit MNSQ and Outfit MNSQ, and were removed from the scale (see Table 3). Reanalysis indicated that the remaining items possessed satisfactory Infit and Outfit MNSQ between 0.6 and 1.4 [56] inferring a lack of 'noise' or randomness within the measure, and no substantial DIF across gender (Table 3). Acceptable item reliability and person reliability (Social Comparison = .92 and .85, Hope = .97 and .91, Negative Expectation = .97 and .89, Futility = .97 and .80) existed for each factor. Moreover, the item separation index and the person separation index (Social Comparison = 3.45 and 2.41, Hope = 6.13 and 3.13, Negative Expectation = 5.89 and 2.83, Futility = 5.27 and 2.06) were all satisfactory, distinguishing high and low ability in the sample [57] and inferring a good spread of items.

**Table 4. Correlations between GHS, GHS subscales and the concepts used to establish their concurrent validity in Study 1.**

| | Variable | |
|---|---|---|
| **Study 1 (*N* = 305)** | **THS** | **RSE** |
| GHS total | .71** | -.46** |
| Social Comparison | .68** | -.45** |
| Hope | -.85** | .47** |
| Negative Expectations | .86** | -.51** |
| Futility | .66** | -.41** |

*Note.* GHS = General Hopelessness Scale, THS = Trait Hopelessness Scale, RSE = Self-esteem;

** $p < .001$

**Validity and reliability.** Concurrent validity was assessed using the THS and the RSE scale (Table 4). The GHS and its subscales correlated strongly with the THS alongside a large association with the RSE. Using alpha and omega coefficients, reliability for the overall scale was good ($\alpha = .94$, $\omega = .95$). Social Comparison evidenced good reliability ($\alpha = .89$, $\omega = .89$). Similarly, Hope was satisfactorily reliable ($\alpha = .87$, $\omega = .89$), in addition to Negative Expectations ($\alpha = .91$, $\omega = .91$), and Futility ($\alpha = .79$, $\omega = .79$).

**Ceiling and floor effects.** Examination of GHS score distributions occurred for floor and ceiling effects. Floor effects represent a limitation of a measure whereby the scale cannot determine decreased performance beyond a specific level. Likewise, ceiling effects indicate the opposite extreme [58]. Following the guidelines of Terwee et al. [59], a floor or ceiling effect existed if 15% or more of the participants reported lowest (i.e., 21) or highest (i.e., 147) possible scores. Analysis of the GHS found negligible floor and ceiling effects. Specifically, the lowest score was 21, with 1 (.3%) possessing this. The highest score was 129, also with 1 participant (.3%) indexing this. Negligible floor and ceiling effects also existed for the four factors.

## Conclusion

Initial analyses identified four underlying factors for the GHS. Further tests of performance revealed that the GHS and its emergent subscales demonstrated satisfactory psychometric properties.

## Study 2 Materials and methods

### Participants

Study 2 consisted of 326 participants (224 women, 68%; 102 men, 32%), mean age 26.52 (*SD* = 8.74), range 18 to 71. Participants were typically from the UK (34%). In terms of vocational status, 74% of the sample were students, 24% employed and 2% unemployed (Table 1). Data screening revealed acceptable univariate skewness and kurtosis for study variables, as with Study 1 (i.e., between -2.0 to +2.0). In addition, non-normality existed (kurtosis $b2p = 26.96$, $p < 0.001$, skewness $b1p = 98.30$, $p < 0.001$).

### Measures

Study 2 consisted of a series of antithetical measures. These included the Positive and Negative Affect Scale [60], the Revised Life Orientation Test [61] and the Adult Hope Scale [62]. These measures were chosen for their adversative nature to hopelessness.

**The Positive and Negative Affect Scale (PANAS).** PANAS consists of two scales measuring positive and negative affect. Respondents indicate the extent to which they usually experience 20 affects (e.g., 'jittery', 'proud'), using a 5-point Likert scale ($1 =$ *very slightly or not at all*, $5 =$ *extremely*). Scores for each scale are achieved by totalling the positive and negatively valanced words. The present study found good reliability for both scales (PA $\alpha = .89$, $\omega = .89$; NA $\alpha = .88$, $\omega = .88$), which is in-line with previous research [60].

**The Revised Life Orientation Test (LOT-R).** The LOT-R is a 10-item measure assessing optimism in the form of general expectancy (e.g., 'In uncertain times, I usually expect the best') using a 5-point Likert scale ($0 =$ *strongly disagree*, $4 =$ *strongly agree*). The present study found good internal consistency ($\alpha = .82$, $\omega = .82$).

**The Adult Hope Scale (AHS).** The AHS is a 12-item measure of a respondent's level of hope, which can be divided into two subscales (1) Agency and (2) Pathways. Each item is answered using an 8-point Likert scale ranging from Definitely False to Definitely True. The present study focused on the total scale, which was acceptably reliable ($\alpha = .79$, $\omega = .83$).

## Procedure

The procedure for Study 2 was identical to Study 1.

## Analytical strategy

CFA (MPlus Version 8.6) examined data-model fit of the superior solution from EFA in an independent sample (Study 2). As with Study 1, due to non-normality and ordinal data, CFA was performed on a polychoric correlation matrix using WLSMV estimation. Satisfactory model fit involved CFI > 0.90, RMSEA < .10, and SRMR < .08.

Measurement invariance was examined by fitting and comparing sequentially nested and increasingly constrained CFA models across gender (men and women). Invariance examined equivalence at the configural (factor structure), metric (factor loadings), and scalar (item intercepts) levels. Invariance was determined by a CFI difference ($\Delta$CFI) $\leq 0.01$ and RMSEA difference ($\Delta$RMSEA) $\leq 0.015$ [63], or a non-significant change in $\chi^2$. Concurrent validity included aligning the GHS with the PANAS, LOT-R and AHS. Cronbach's alpha and coefficient omega ($\omega$) examined internal consistency prior to consideration of ceiling and floor effects.

## Study 2 Results

**Confirmatory Factor Analysis (CFA).** A replication of the four-factor model with Study 2 revealed (using CFA) satisfactory fit, $\chi^2$ (203) = 778.66, $p < .001$, CFI = .94, RMSEA = .09 (90% CI of .08 to .10), SRMR = .05. Inspection of standardized parameter estimates indicated that all items loaded above .5 [64] apart from item 9 (loading of .41). The four factors reported large inter-factor correlations with one another (Fig 1).

**Measurement invariance.** Multi-group analysis comparing men and women supported metric invariance based on the change criteria in fit statistics specified a priori (Table 5): Model 1 vs. Model 2 ($\Delta$CFI = .006, $\Delta$RMSEA = .004). Support also existed for scalar invariance: Model 2 vs. Model 3 ($\Delta$CFI = .004, $\Delta$RMSEA = .014).

**Validity and reliability.** Moderate to large correlations occurred between the GHS and its subscales with the PANAS subscales and AHS (Table 6). Large negative correlations existed with the LOT-R measure. The Hope subscale expressed a similarly divergent pattern as witnessed in Study 1, signifying divergent validity. Reliability was good for the overall scale ($\alpha = .93$, $\omega = .93$), Social Comparison ($\alpha = .82$, $\omega = .83$), Hope ($\alpha = .84$, $\omega = .85$), Negative Expectations ($\alpha = .89$, $\omega = .89$), and Futility ($\alpha = .75$, $\omega = .75$).

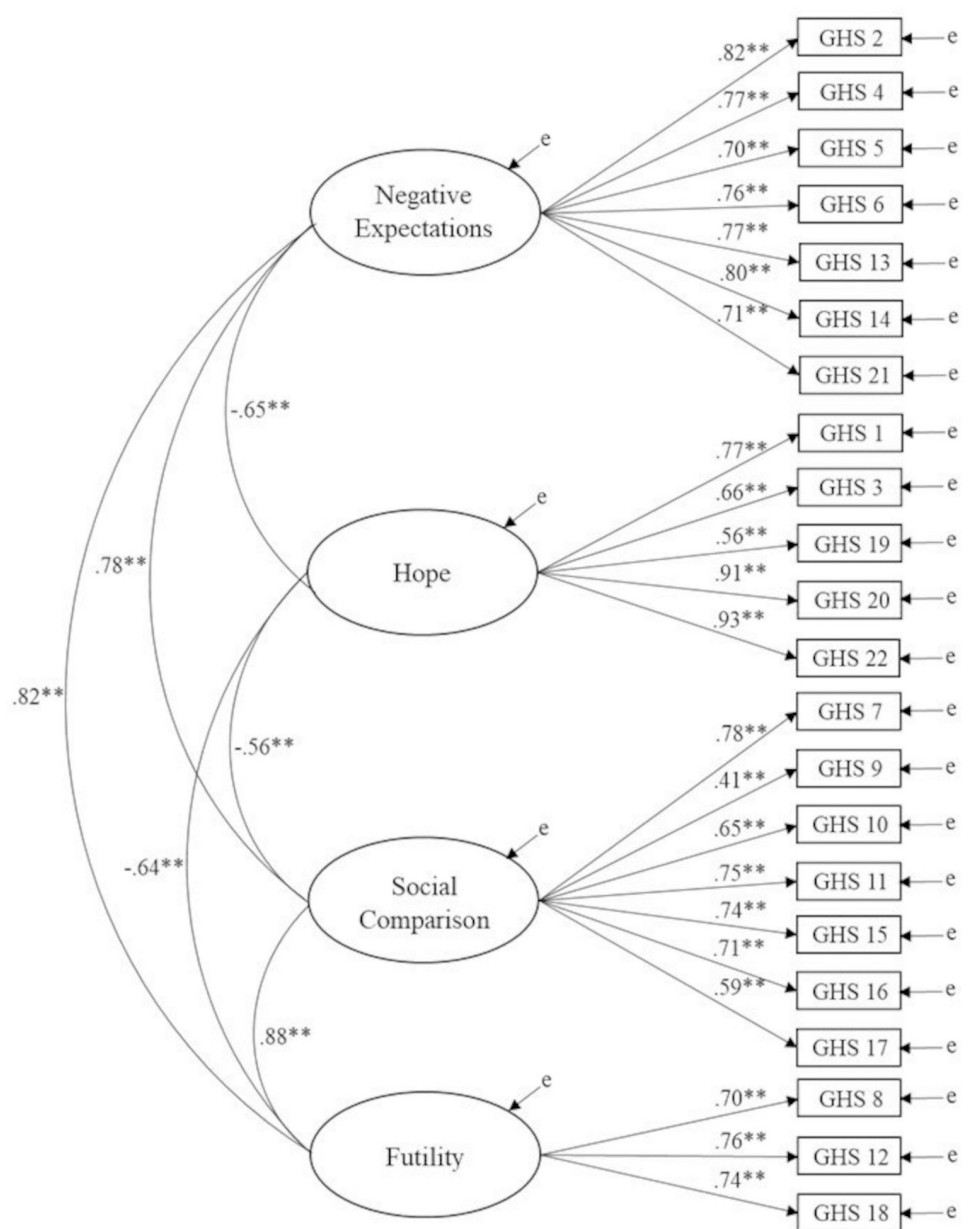

**Fig 1. Factor structure of the General Hopelessness Scale.** *Note.* Ellipses represent latent variables; measured variables are represented by rectangles; error is not shown but was specified for all variables. ** *p* < .001.

**Table 5. Measurement invariance for the General Hopelessness Scale by gender for Study 2.**

| Model | $\chi^2$ | *df* | CFI | SRMR | RMSEA (90% CI) | $\Delta\chi^2$(df) | ΔCFI | ΔRMSEA |
|---|---|---|---|---|---|---|---|---|
| Model 1 (Equal Form, Configural) | 975.43** | 406 | .94 | .05 | .09 (.08-.10) | - | - | - |
| Model 2 (Equal loadings, Metric) | 973.72** | 424 | .94 | .05 | .08 (.08-.09) | 22.11 (18) | .006 | .004 |
| Model 3 (Equal loadings and intercepts, Scalar) | 1020.32** | 530 | .95 | .05 | .07 (.06-.08) | 141.84 (106) | .004 | .014 |

Note.

** indicates *p* < .001

Table 6. Correlations between GHS, GHS subscales and the concepts used to establish their concurrent validity in Study 2.

| Study 2 (*N* = 326) | Variable | | | |
|---|---|---|---|---|
| | **PANAS-P** | **PANAS-N** | **LOT-R** | **AHS** |
| GHS total | -.38** | .53** | -.64** | -.33** |
| Social Comparison | -.49** | .55** | -.59** | -.37** |
| Hope | .66** | -.30** | .61** | .72** |
| Negative Expectations | -.42** | .47** | -.78** | -.46** |
| Futility | -.50** | .40** | -.52** | -.51** |

*Note*. GHS = General Hopelessness Scale, PANAS-P = Positive Affect, PANAS-N = Negative affect, LOT-R = Optimism, AHS = Hope;

** *p* < .001

**Ceiling and floor effects.** The lowest score (31) was greater than in Study 1, with only 1 participant reporting this (.3%) and the highest score was 140, again for 1 participant (.3%). Consistent with Study 1, negligible floor and ceiling effects existed for the four factors.

## Conclusion

The four-factor model demonstrated satisfactory model fit and invariance when tested on an independent sample. Consistent with Study 1, the GHS and its subscales exhibited satisfactory psychometric qualities in terms of validity, reliability, and ceiling and floor effects.

## Discussion

Based on Beck's cognitive theory of depression [65] and Abramson's theory of hopelessness depression [36], this project developed a novel, psychometrically sound measure of hopelessness for use with general samples. This was necessary because attempts to reproduce the BHS factor structure reported by Beck et al. [1] have produced inconsistent results [e.g., 12, 15, 18, 23, 30]. Additionally, theorists have raised concerns about the ability of the BHS to discriminate between lower levels of hopelessness such as those found in non-clinical samples or improving patients [2, 30]. Indeed, this was only possible when the factor structure was altered to produce a simpler model [11, 17]. Finally, adoption of an integrated approach that combines Beck's cognitive theory of depression [65] and Abramson's theory of hopelessness depression [36] opens the possibility for more effective therapeutic interventions in which techniques from different cognitive theories are optimally amalgamated [41]. In this context, the GHS will potentially extend understanding of hopelessness in non-clinical populations and help to facilitate the development of conceptually integrated assessments.

Regarding analysis, EFA identified four factors, which approximated the theoretical underpinnings of items during its construction. Specifically, Negative Expectations, Hope, Social Comparison, and Futility. Rasch analysis and CFA consolidated these factors by informing the removal of underperforming and/or inconsistent items. This, reduced the number of items to 21, strengthened model fit, and improved inter-factor covariance. A concomitant advantage of a shorter emergent measure being enhanced practicality (easier to complete and comprehend). Whilst reliabilities, both for specific factors and overall measure, were good, future research is necessary to establish its efficacy. Nonetheless, the use of a non-clinical sample in the GHS development and the observed concurrent validity with established measures indicated promising scale performance.

Additionally, the study provided information about ceiling and floor effects. These are viewed as problematic when greater than 15% of a sample either produce the highest or lowest

available score [59]. Effects of this magnitude did not exist in this study, suggesting that the GHS adequately captured general hopelessness scores. This is a criticism, which scholars have levelled at the BHS [17, 18, 27]. Additional tests of invariance demonstrated satisfactory stability of factor structure, loadings, and intercepts across gender. Thus, analysis specified that the GHS was a psychometrically sound measure of hopelessness.

Conceptually, the synergy of key elements of Beck's cognitive model of hopelessness [1] and Abramson's theory of hopelessness depression [36] combined to reflect core characteristics of hopelessness. Specifically, negative, dysfunctional perceptions and a cynical view of the future. Moreover, the GHS assessed these features concomitant with other aspects that contribute towards hopelessness, such as cognitive errors that occur when comparing perceptions of one's own ability to the seemingly unattainable competence of others. The inclusion of facets of social comparison and futility ensured that the GHS sampled a breadth of construct content.

The inclusion of social comparison stemmed from Abramson's [36] examples of how hopelessness depression develops. Hence, it encapsulated production of cognitive errors and Beck's cognitive triad and was commensurate with relationships between social comparison and self-esteem. Particularly, the observation that those who feel comfortable in their perceptions of themselves, in comparison to others, possess positive self-esteem [66]. Accordingly, negative social comparisons can adversely affect self-esteem [67–69] and serve as a precursor of depression, as well as hopelessness. Subsequent studies with the GHS should explore the benefit of this factor.

EFA resulted in the emergence of a nuanced three-item motivation factor "futility". This reflected aversion to partake in/or maintain behaviours/habits. Explicitly, perceived personal inadequacy and lack/absence of initiative. This interpretation was consistent with the delineation of futility as low self-efficacy and heightened fear of failure (e.g., "*I feel that giving up is easier than failing*" and "*I avoid attempting new things in case I find them difficult*"). This emphasis differed from the BHS motivational factor, which focuses on loss of motivation and poor future expectations. Thus, conceptually futility corresponded with negative self-views (cognitive triad) and self-referential style (hopelessness theory). Explicitly, undesirable personal evaluation and internalisation of failure result in lack of motivation and volition [70, 71].

## Limitations

Although this study reported that the GHS possessed good psychometric properties it is important to note limitations. One concern was conceptual distinctiveness. This was potentially an issue because of similarities between hopelessness (i.e., the propensity to overestimate adverse events and underestimate the probability of positive occurrences) [3] and locus of control (LOC) (i.e., individual beliefs about control of life events). Accordingly, Rotter [72] defines LOC as the extent to which an individual perceives an outcome as dependent on their own actions or external forces; this is assessed on a continuum from internal to external orientation [73]. Internal LOC views control as personal, whereas external LOC attributes control to external factors. Thus, LOC reflects generalized confidence. Explicitly, experience of the self as either a causal agent (optimism/hope), or a victim of circumstance (pessimism/hopelessness) [74].

In the context of the present paper, Levenson's [75] reconceptualization of external locus is highly relevant. Levenson divided LOC into control by powerful others, and control by chance or fate. These loci of control represent can be interpreted as fatalism and as contributing to hopelessness [74]. Despite apparent similarities between hopelessness and LOC, it is important to remember that constructs of perceived control (including also self-efficacy [76], learned

helplessness [77], and causal attributions [78]) derive from distinct conceptual backgrounds [74]. Correspondingly, studies typically report only moderate positive correlations between hopelessness and LOC [e.g., 79, 80]. Although, the usefulness of cross study comparisons is restricted by investigators' use of different measurement and samples, it is evident that the two constructs are distinct and reflect different cognitive-perceptual processes.

Explicitly, external LOC denotes a general thinking style that captures elements of hopelessness, which is an extreme reaction to adverse circumstances. Commensurate with this interpretation, Lefcourt [81] drawing on Richter [82], delineates hopelessness as giving up when all avenues of escape appear to be closed and the future holds no hope. Furthermore, consistent with the dimensions identified in this paper, hopelessness embodies pessimism about the future (lack of hope and negative expectations) and resignation to the ineffectiveness of changing the future (futility) [83]. The GHS suggests that these judgments derive from both internalised representations of the self and comparisons with others (social comparison). To extend scholarly understanding and assess the validity of these assumptions future research with the GHS should explore relationships between dimensions of hopelessness and LOC.

A limitation of the present study was the failure to concurrently evaluate the GHS within a clinical sample. Future research should undertake this comparison to ensure that the GHS performs well across different settings. Additionally, assessment of the GHS alongside the BHS is required. This was not possible in the current report due to financial restrictions and sampling constraints (i.e., lack of access to clinical sample). This would establish that the GHS (vs. BHS) is more able to discriminate hopelessness scores in the low to moderate response range. This could also inform the development of more sensitive interventions and recovery monitoring strategies.

Though the present paper demonstrated the validity and internal reliability of the GHS, further research is required to verify the measure's external reliability (i.e., temporal stability). Hence, subsequent psychometric evaluation should assess the test-retest reliability of the GHS. This is important because external reliability shows that participant scores are consistent and reproduced under comparable conditions on independent occasions [84]. Hence, test-retest reliability is integral to scale performance since it ensures the effective measurement of hopelessness over time.

Moreover, test-retest reliability complements validity by verifying measurement stability. Consequently, in the context of mental health-related outcomes test-retest reliability is necessary to the assessment of treatment and intervention efficacy. A vital consideration when establishing test-retest is the gap between scale completions. The interval needs to be sufficient to demonstrate adequate stability, but not too long for substantial attrition to occur, or allow extrinsic factors to influence scores. Hence with health measures such as the GHS, intervals of 1–2 weeks are typically employed [85]. Although, interval length is often constrained by practicality and subsequently, validation papers often use durations of opportunity [86]. Regardless of period between scale completions, establishing test-retest will further enhance the psychometric properties of the GHS. This is especially necessary since reliability across multiple times, contexts, and users is generally poorly reported within psychological literature [87].

In terms of measurement development, future research could produce an abridged version of the GHS (i.e., 1–3 items per dimension) for use in big panel surveys. Within large test batteries brevity increases end-user accessibility, by reducing survey length, concomitantly facilitating engagement. Additionally, shorter, refined measures reduce the cognitive load placed on respondents and are accordingly less likely to result in fatigue. The development of a short version of the GHS, however, is a long iterative process that should only occur after the measure is thoroughly evaluated and refined. Particularly, it is important to ensure that any emergent scale adequately samples construct breadth.

## Acknowledgments

We would like to thank the participants of the study, without whom this study would not have been possible.

## Author Contributions

**Conceptualization:** Ken Drinkwater, Andrew Denovan, Neil Dagnall, Chris Williams.

**Data curation:** Andrew Denovan, Neil Dagnall, Chris Williams.

**Formal analysis:** Andrew Denovan, Neil Dagnall.

**Investigation:** Ken Drinkwater, Andrew Denovan, Neil Dagnall.

**Methodology:** Ken Drinkwater, Andrew Denovan, Neil Dagnall, Chris Williams.

**Writing – original draft:** Andrew Denovan, Neil Dagnall, Chris Williams.

**Writing – review & editing:** Ken Drinkwater, Andrew Denovan, Neil Dagnall.

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
