## [Decision Letter · Decision Letter 0]

17 Jan 2023

PONE-D-22-15026The General Hopelessness Scale: Development of a measure for non-clinical samples PLOS ONE

Dear Dr. Ken Drinkwater,

Thank you for submitting your manuscript to PLOS ONE. After careful consideration, we feel that it has merit but does not fully meet PLOS ONE’s publication criteria as it currently stands. Therefore, we invite you to submit a revised version of the manuscript that addresses the points raised during the review process.

The manuscript is relevant and presents a new psychometric measure of hopelessness.

However, to accept the manuscript, it is recommended to consider the adjustments requested by the reviewers and, here, endorsed.

First, a much better way to guide your research is to address a scientific problem related to a relevant problem. Successful manuscripts usually answer at least one of the following questions: Is it a solution to an established and relevant problem? Is it an extension of a well-known technique? Is your contribution improving a technique or methodology? In any case, the literature review must clearly explain where your contribution fits.

Second, the abstract needs to present the participants, specifying pertinent characteristics, such as age, sex, and ethnic and/or racial group; the essential characteristics of the study method - particularly those most likely to be used in electronic searches; the main results, including effect sizes and confidence intervals and/or levels of statistical significance; and conclusions and implications or applications.

Third, for the methods and results sections, please refer to the APA recommendations. For more details, see: Appelbaum, M., Cooper, H., Kline, R. B., Mayo-Wilson, E., Nezu, A. M., & Rao, S. M. (2018). Journal article reporting standards for quantitative research in psychology: The APA Publications and Communications Board task force report. *American Psychologist*, *73*(1), 3–25. https://doi.org/10.1037/amp0000191.

Also, consider updating the reference for the cutoffs used for CFA. For more details, see: van Laar, S., & Braeken, J. (2022). Caught off Base: A Note on the Interpretation of Incremental Fit Indices. *Structural Equation Modeling: A Multidisciplinary Journal*, *29*(6), 935–943. https://doi.org/10.1080/10705511.2022.2050730.

Four, for data analysis and results.

1. Confirmatory Factor Analysis (CFA). Due to the ordinal nature of dataset and non-normal distribution, it is recommended to use a Weighted Least Squares Mean and Variance-Adjusted (WLSMV) estimator with polychoric matrices for performing the CFA. For more details, see: DiStefano, C., & Morgan, G. B. (2014). A Comparison of Diagonal Weighted Least Squares Robust Estimation Techniques for Ordinal Data. *Structural Equation Modeling: A Multidisciplinary Journal*, *21*(3), 425–438. https://doi.org/10.1080/10705511.2014.915373

2. Rasch analysis. Due to the ordinal nature of the data set, it is necessary to fit the analysis to the gradation response model. Is a mathematical model family that deals with ordered polytomous categories. These ordered categories include ratings such as strongly disagree, disagree, agree, and strongly agree, which are used in the manuscript. For more details, see: Samejima, F. (2016). Graded response models. In *Handbook of item response theory, volume one* (pp. 123-136). Chapman and Hall/CRC. Furthermore, it is necessary to consider the discrimination ("a" parameter). Because the goal of most psychometric measures is to distinguish between examinees, items with higher discriminations ("a" parameter) are frequently regarded as better. Items with very low parameters provide minimal measurement information and might be considered for change or removal.

Finally, once you have revised the manuscript, check it. Are all fonts visible, particularly in equations? Is numbering consistent across sections, figures, tables, etc.? Are all references, figures, and tables cited? Are all the references listedcited? Are all the figures and tables cited?

We look forward to receiving your revised manuscript.

Kind regards,

Jorge Artur Peçanha de Miranda Coelho

Academic Editor

PLOS ONE

Journal Requirements:

2. Please change "female” or "male" to "woman” or "man" as appropriate, when used as a noun (see for instance https://apastyle.apa.org/style-grammar-guidelines/bias-free-language/gender)."

3. Peer review at PLOS ONE is not double-blinded (https://journals.plos.org/plosone/s/editorial-and-peer-review-process). For this reason, authors should include in the revised manuscript all the information removed for blind review.

4. Please provide further information on how participants were selected/recruited"

5. Please provide additional details regarding participant consent. In the ethics statement in the Methods and online submission information, please ensure that you have specified what type you obtained (for instance, written or verbal, and if verbal, how it was documented and witnessed). If your study included minors, state whether you obtained consent from parents or guardians. If the need for consent was waived by the ethics committee, please include this information

Reviewers' comments:

Reviewer's Responses to Questions

**Comments to the Author**

1. Is the manuscript technically sound, and do the data support the conclusions?

Reviewer #1: Yes

Reviewer #2: Yes

2. Has the statistical analysis been performed appropriately and rigorously? 

Reviewer #1: Yes

Reviewer #2: No

3. Have the authors made all data underlying the findings in their manuscript fully available?

Reviewer #1: Yes

Reviewer #2: Yes

4. Is the manuscript presented in an intelligible fashion and written in standard English?

Reviewer #1: Yes

Reviewer #2: Yes

5. Review Comments to the Author

Reviewer #1: In the manuscript "The General Hopelessness Scale: Development of a measure for non-clinical samples", the authors developed a new instrument to measure hopelessnes, aming to better discriminate between lower levels of hopelessness.

In general the topic of this manuscript is interesting for readers of this journal, the analyses are according to high standards, and the paper is written clear and nicely.

I have only two remarks:

- It would be good to also develop a short scale with 1-3 items per dimension to allow an effective and quick measurement in big panel surveys.

- Maybe an unpopular opinion and also not really constructive, but does the (scientific) world really need a measure of hopelessness? We have pretty good (and short) screening intruments for depression. Locus of control is also a very similar concept. So, what is the major advantage of measuring hopelessness? Why do we need to include this in our surveys?

Signed,

David Richter.

Reviewer #2: The manuscript reviewed here aimed at presenting the development of a novel measure of hopelesness, based on Beck's cognitive theory of depression, that could be used to non-clinical samples.

I do believe the studies presented are interesting and novel, showing a promising measure that could be used for research and also for therapeutic purposes, in order to understand an individual's beliefs regarding their own level of hope / hopelesness.

However, I also believe the manuscript is not ready for publication, as it needs further development in the introduction and to change the softwares used in the analyses.

Please, see my comments below.

ABSTRACT

* It doesn't present sample information. The abstract should contain information regarding sample, instruments and results.

INTRODUCTION

* Introduction is short on presenting the main concept of the paper, which is Hopelesness. Only one paragraph presents something about it before presenting the BHS. I believe a few more paragraphs explaining its importance and nomological network might be useful to locate the reader regarding the current study of the topic.

* The authors make interesting comments on the different structures observed in the literature and the limitations of using the BHS in non-clinical populations.

* The paper also justifies accordingly the need to develop a non-clinical hopelesness instrument, for which I commend the authors.

METHODS

* Participants and instruments are presented objectively, with all necessary information.

However, there is no information on how the study was presented to potential participants or how they came to know about the study. Was there any inclusion / exclusion criteria for participant selection? Procedures paragraph should include this information.

RESULTS

* I do believe the most recent algorithms for conducting a proper EFA are included in the FACTOR software, especially when involving the discussion regarding items on psychological instruments being ordinal and not scalar items. In this case, I suggest the authors change the software used in the EFA from SPSS to FACTOR.

* Also regarding the CFA, the AMOS software also doesn't include the most recent algorithms for these analyses. Softwares such as JASP, JAMOVI or the R language are much improved when compared to AMOS.

* For both comments here regarding the softwares, there might be no difference in the results, but we would be sure the analyses were conducted using the best softwares we have nowadays.

* Table 2 doesn't present the results the best way possible. We need to see all factor loadings in all dimensions to visualize its distribution.

* Why the term futility was used to describe "an individual's aversion to initiate or maintain new behaviours"? In current language, the term futility would be describing something trivial or pointless. I do understand the term emphasizes the individual's belief that changing or maintaning change is ineffective or fruitless. However, the definition is "aversion to initiate or maintain new behaviors". For me, the emphasis in this sentence is the aversion and not what the individual uses to justify their own aversion (the pointlessness of trying). This is only to think about as the terms we choose to describe the dimensions are important as they carry the meaning of the psychological construct underlying the items.

* Also, regarding the meaning of the items presented in Table 2 and their attributed dimension, I have a few doubts regarding their belonging to that definition. To exemplify, see below:

- Item 7 describes everything as pointless. Why is this item in the Social Comparison dimension and not in the Futility dimension, like the item 8?

- Item 9 is a clear case of Social Comparison, but why does the item 10 belong in the same dimension? Where is the social comparison in this case? Same question apply for items 11 and 16.

DISCUSSION

* I see a little bit of a discussion regarding the term 'futility' in the discussion session, which is very interesting. However, my comment remains on why calling it 'Futility' and not "Aversion to change" or "Lack of intrinsic motivation"?

* A more detailed paragraph on future studies regarding the potential use of the new measure is also lacking.

6. PLOS authors have the option to publish the peer review history of their article (what does this mean?). If published, this will include your full peer review and any attached files.

Reviewer #1: **Yes: **David Richter

Reviewer #2: No

---

## [Author Response · Author response to Decision Letter 0]

27 Feb 2023

PONE-D-22-15026

The General Hopelessness Scale: Development of a measure for non-clinical samples 

PLOS ONE

Dear Dr. Ken Drinkwater,

Thank you for submitting your manuscript to PLOS ONE. After careful consideration, we feel that it has merit but does not fully meet PLOS ONE’s publication criteria as it currently stands. Therefore, we invite you to submit a revised version of the manuscript that addresses the points raised during the review process.

The manuscript is relevant and presents a new psychometric measure of hopelessness.

However, to accept the manuscript, it is recommended to consider the adjustments requested by the reviewers and, here, endorsed.

Comment:

First, a much better way to guide your research is to address a scientific problem related to a relevant problem. Successful manuscripts usually answer at least one of the following questions: Is it a solution to an established and relevant problem? Is it an extension of a well-known technique? Is your contribution improving a technique or methodology? In any case, the literature review must clearly explain where your contribution fits.

Response:

Thank you for this comment. We have included an advanced organiser at the beginning of the Introduction. This frames the purpose of the paper. We have also revisited the present study section to add greater continuity and expanded on the Discussion. This enhances the structure and flow of the paper around the research question.

Comment:

Second, the abstract needs to present the participants, specifying pertinent characteristics, such as age, sex, and ethnic and/or racial group; the essential characteristics of the study method - particularly those most likely to be used in electronic searches; the main results, including effect sizes and confidence intervals and/or levels of statistical significance; and conclusions and implications or applications.

Response:

Thank you. We have added information on the sample(s), assessed constructs, and results.

Comment:

Third, for the methods and results sections, please refer to the APA recommendations. For more details, see: Appelbaum, M., Cooper, H., Kline, R. B., Mayo-Wilson, E., Nezu, A. M., & Rao, S. M. (2018). Journal article reporting standards for quantitative research in psychology: The APA Publications and Communications Board task force report. American Psychologist, 73(1), 3–25. https://doi.org/10.1037/amp0000191.

Response:

Thank you. We have added additional analyses in the paper and improved clarity. In addition, we have restructured the method and results to improve clarity and organisation for the reader.

 

Comment:

Also, consider updating the reference for the cutoffs used for CFA. For more details, see: van Laar, S., & Braeken, J. (2022). Caught off Base: A Note on the Interpretation of Incremental Fit Indices. Structural Equation Modeling: A Multidisciplinary Journal, 29(6), 935–943. https://doi.org/10.1080/10705511.2022.2050730.

Response:

Thank you. We have updated the reference and used Rex Kline (2015) and Wang and Wang (2012). These indices are cited by papers published in the recent period of 2021 to 2023:

Archibald, D. E., Graham, C. R., & Larsen, R. (2021). Validating a blended teaching readiness instrument for primary/secondary preservice teachers. British Journal of Educational Technology, 52(2), 536-551.

McAninch, K. G., Basinger, E. D., Delaney, A. L., & Wehrman, E. C. (2023). Integrating relational turbulence theory and the theory of resilience and relational load to investigate the relationships of couples with chronic illness. Communication Quarterly, 71(1), 1-21.

Comment:

Four, for data analysis and results.

1. Confirmatory Factor Analysis (CFA). Due to the ordinal nature of dataset and non-normal distribution, it is recommended to use a Weighted Least Squares Mean and Variance-Adjusted (WLSMV) estimator with polychoric matrices for performing the CFA. For more details, see: DiStefano, C., & Morgan, G. B. (2014). A Comparison of Diagonal Weighted Least Squares Robust Estimation Techniques for Ordinal Data. Structural Equation Modeling: A Multidisciplinary Journal, 21(3), 425–438. https://doi.org/10.1080/10705511.2014.915373

Response:

Thank you. We have reanalysed the data using WLSMV with polychoric matrices, as suggested. In addition, we used this estimator for the EFA.

Comment:

2. Rasch analysis. Due to the ordinal nature of the data set, it is necessary to fit the analysis to the gradation response model. Is a mathematical model family that deals with ordered polytomous categories. These ordered categories include ratings such as strongly disagree, disagree, agree, and strongly agree, which are used in the manuscript. For more details, see: Samejima, F. (2016). Graded response models. In Handbook of item response theory, volume one (pp. 123-136). Chapman and Hall/CRC. Furthermore, it is necessary to consider the discrimination ("a" parameter). Because the goal of most psychometric measures is to distinguish between examinees, items with higher discriminations ("a" parameter) are frequently regarded as better. Items with very low parameters provide minimal measurement information and might be considered for change or removal.

Response:

Thank you. We do not have access to software for performing this analysis, and we are unfamiliar with the technique. However, please note that we used the Rasch Rating Scale Model (RSM), which has been demonstrated to be effective/appropriate for use with Likert-scale surveys (comprising categories from, e.g., strongly disagree to strongly agree) and data considered to be ordinal. See Yamashita (2022) for an exposition.

Yamashita, T. (2022). Analyzing Likert scale surveys with Rasch models. Research methods in applied linguistics, 1(3), 100022. https://doi.org/10.1016/j.rmal.2022.100022

Finally, once you have revised the manuscript, check it. Are all fonts visible, particularly in equations? Is numbering consistent across sections, figures, tables, etc.? Are all references, figures, and tables cited? Are all the references listedcited? Are all the figures and tables cited?

We look forward to receiving your revised manuscript.

Kind regards,

Jorge Artur Peçanha de Miranda Coelho

Academic Editor

PLOS ONE

Journal Requirements:

2. Please change "female” or "male" to "woman” or "man" as appropriate, when used as a noun (see for instance https://apastyle.apa.org/style-grammar-guidelines/bias-free-language/gender)."

3. Peer review at PLOS ONE is not double-blinded (https://journals.plos.org/plosone/s/editorial-and-peer-review-process). For this reason, authors should include in the revised manuscript all the information removed for blind review.

4. Please provide further information on how participants were selected/recruited"

5. Please provide additional details regarding participant consent. In the ethics statement in the Methods and online submission information, please ensure that you have specified what type you obtained (for instance, written or verbal, and if verbal, how it was documented and witnessed). If your study included minors, state whether you obtained consent from parents or guardians. If the need for consent was waived by the ethics committee, please include this information

Reviewers' comments:

Reviewer's Responses to Questions

Comments to the Author

1. Is the manuscript technically sound, and do the data support the conclusions?

Reviewer #1: Yes

Reviewer #2: Yes

2. Has the statistical analysis been performed appropriately and rigorously?

 Reviewer #1: Yes

Reviewer #2: No

3. Have the authors made all data underlying the findings in their manuscript fully available?

Reviewer #1: Yes

Reviewer #2: Yes

4. Is the manuscript presented in an intelligible fashion and written in standard English?

Reviewer #1: Yes

Reviewer #2: Yes

5. Review Comments to the Author

Reviewer #1: In the manuscript "The General Hopelessness Scale: Development of a measure for non-clinical samples", the authors developed a new instrument to measure hopelessnes, aming to better discriminate between lower levels of hopelessness.

In general the topic of this manuscript is interesting for readers of this journal, the analyses are according to high standards, and the paper is written clear and nicely.

Comment:

I have only two remarks:

- It would be good to also develop a short scale with 1-3 items per dimension to allow an effective and quick measurement in big panel surveys.

Response:

Thank you. We have included a section on the importance of developing an abridged version of the measure in the Limitations section. We did not develop this within the paper because this is a long, iterative process, and it is important to firstly assess the initial measure.

Comment:

- Maybe an unpopular opinion and also not really constructive, but does the (scientific) world really need a measure of hopelessness? We have pretty good (and short) screening intruments for depression. Locus of control is also a very similar concept. So, what is the major advantage of measuring hopelessness? Why do we need to include this in our surveys?

Response:

Thank you. We have included greater discussion of the similarities and differences between hopelessness and LOC construct within the Discussion.

Signed,

David Richter.

Reviewer #2: The manuscript reviewed here aimed at presenting the development of a novel measure of hopelesness, based on Beck's cognitive theory of depression, that could be used to non-clinical samples.

I do believe the studies presented are interesting and novel, showing a promising measure that could be used for research and also for therapeutic purposes, in order to understand an individual's beliefs regarding their own level of hope / hopelesness.

However, I also believe the manuscript is not ready for publication, as it needs further development in the introduction and to change the softwares used in the analyses.

Please, see my comments below.

ABSTRACT

Comment:

* It doesn't present sample information. The abstract should contain information regarding sample, instruments and results.

Response:

Thank you. We have added information on the sample(s), assessed constructs, and results.

 

INTRODUCTION

Comment:

* Introduction is short on presenting the main concept of the paper, which is Hopelesness. Only one paragraph presents something about it before presenting the BHS. I believe a few more paragraphs explaining its importance and nomological network might be useful to locate the reader regarding the current study of the topic.

Response:

Additional content has been added.

* The authors make interesting comments on the different structures observed in the literature and the limitations of using the BHS in non-clinical populations.

* The paper also justifies accordingly the need to develop a non-clinical hopelesness instrument, for which I commend the authors.

METHODS

Comment:

* Participants and instruments are presented objectively, with all necessary information.

However, there is no information on how the study was presented to potential participants or how they came to know about the study. Was there any inclusion / exclusion criteria for participant selection? Procedures paragraph should include this information.

Response:

Thank you. Content has been added.

RESULTS

Comment:

* I do believe the most recent algorithms for conducting a proper EFA are included in the FACTOR software, especially when involving the discussion regarding items on psychological instruments being ordinal and not scalar items. In this case, I suggest the authors change the software used in the EFA from SPSS to FACTOR.

Response:

Thank you for this suggestion. In response to the suggestion of the Editor and yourself, we changed the analysis program and opted to use MPlus with WLSMV estimation. MPlus comprises an excellent algorithm.

Comment:

* Also regarding the CFA, the AMOS software also doesn't include the most recent algorithms for these analyses. Softwares such as JASP, JAMOVI or the R language are much improved when compared to AMOS.

Response:

Thank you. In response to the Editor and your recommendations, we opted to use MPlus with WLSMV estimation.

 

Comment:

* For both comments here regarding the softwares, there might be no difference in the results, but we would be sure the analyses were conducted using the best softwares we have nowadays.

Response:

Thank you. The results were similar, despite using a different estimator.

Comment:

* Table 2 doesn't present the results the best way possible. We need to see all factor loadings in all dimensions to visualize its distribution.

Response:

Thank you. We have presented the table to display this information.

Comment:

* Why the term futility was used to describe "an individual's aversion to initiate or maintain new behaviours"? In current language, the term futility would be describing something trivial or pointless. I do understand the term emphasizes the individual's belief that changing or maintaning change is ineffective or fruitless. However, the definition is "aversion to initiate or maintain new behaviors". For me, the emphasis in this sentence is the aversion and not what the individual uses to justify their own aversion (the pointlessness of trying). This is only to think about as the terms we choose to describe the dimensions are important as they carry the meaning of the psychological construct underlying the items.

* Also, regarding the meaning of the items presented in Table 2 and their attributed dimension, I have a few doubts regarding their belonging to that definition. To exemplify, see below:

- Item 7 describes everything as pointless. Why is this item in the Social Comparison dimension and not in the Futility dimension, like the item 8?

- Item 9 is a clear case of Social Comparison, but why does the item 10 belong in the same dimension? Where is the social comparison in this case? Same question apply for items 11 and 16.

Response:

Additional information added to Discussion. 

DISCUSSION

Comment:

* I see a little bit of a discussion regarding the term 'futility' in the discussion session, which is very interesting. However, my comment remains on why calling it 'Futility' and not "Aversion to change" or "Lack of intrinsic motivation"?

Response:

Thank you. This decision was made because the items, more than anything, gave the impression of pointlessness. Also, as added to the Introduction, futility is a manifestation of hopelessness. We are aware that the labelling of factors is a subjective process, and we appreciate your input/insight.

Comment:

* A more detailed paragraph on future studies regarding the potential use of the new measure is also lacking.

Response:

Thank you. We have expanded the section on limitations and suggestions for future research considerably, incorporating further discussion of avenues of future research pertaining to the measure.

6. PLOS authors have the option to publish the peer review history of their article (what does this mean?). If published, this will include your full peer review and any attached files.

Do you want your identity to be public for this peer review? For information about this choice, including consent withdrawal, please see our Privacy Policy.

Reviewer #1: Yes: David Richter

Reviewer #2: No

---

## [Decision Letter · Decision Letter 1]

30 May 2023

The General Hopelessness Scale: Development of a measure for non-clinical samples

PONE-D-22-15026R1

Dear Dr. Drinkwater,

We’re pleased to inform you that your manuscript has been judged scientifically suitable for publication and will be formally accepted for publication once it meets all outstanding technical requirements.

Kind regards,

Karl Bang Christensen, Ph.D.

Academic Editor

PLOS ONE

Additional Editor Comments (optional):

Thank you for revising

Reviewers' comments:

Reviewer's Responses to Questions

**Comments to the Author**

1. If the authors have adequately addressed your comments raised in a previous round of review and you feel that this manuscript is now acceptable for publication, you may indicate that here to bypass the “Comments to the Author” section, enter your conflict of interest statement in the “Confidential to Editor” section, and submit your "Accept" recommendation.

Reviewer #1: All comments have been addressed

Reviewer #2: All comments have been addressed

2. Is the manuscript technically sound, and do the data support the conclusions?

Reviewer #1: Yes

Reviewer #2: Yes

3. Has the statistical analysis been performed appropriately and rigorously? 

Reviewer #1: Yes

Reviewer #2: Yes

4. Have the authors made all data underlying the findings in their manuscript fully available?

Reviewer #1: Yes

Reviewer #2: Yes

5. Is the manuscript presented in an intelligible fashion and written in standard English?

Reviewer #1: Yes

Reviewer #2: Yes

6. Review Comments to the Author

Reviewer #1: The authors did a fine job with the revisionand addressed all my comments. At this point, I have no futher comments.

Reviewer #2: The new revised version of the paper adressed all my comments from the previous round. The only thing missing for me is the Figure 1, which did not appear in my pdf version of the manuscript. However, it is possible to infer its quality, considering the main numbers are described in the text.

7. PLOS authors have the option to publish the peer review history of their article (what does this mean?). If published, this will include your full peer review and any attached files.

Reviewer #1: No

Reviewer #2: No

---

## [Editor Report · Acceptance letter]

13 Jun 2023

PONE-D-22-15026R1 

The General Hopelessness Scale: Development of a measure of hopelessness for non-clinical samples 

Dear Dr. Drinkwater:

I'm pleased to inform you that your manuscript has been deemed suitable for publication in PLOS ONE. Congratulations! Your manuscript is now with our production department. 

Kind regards, 

on behalf of

Dr. Karl Bang Christensen 

Academic Editor

PLOS ONE